# A Data Perspective on Enhanced Identity Preservation for Diffusion Personalization

## Abstract

Large text-to-image models have revolutionized the ability to generate imagery using natural language. However, particularly unique or personal visual concepts, such as your pet, an object in your house, etc., will not be captured by the original model. This has led to interest in how to inject new visual concepts, bound to a new text token, using as few as 4-6 examples. Despite significant progress, this task remains a formidable challenge, particularly in preserving the subject's identity. While most researchers attempt to to address this issue by modifying model architectures, our approach takes a data-centric perspective, advocating the modification of data rather than the model itself. We introduce a novel regularization dataset generation strategy on both the text and image level; demonstrating the importance of a rich and structured regularization dataset (automatically generated) to prevent losing text coherence and better identity preservation. The better quality is enabled by allowing up to 5x more fine-tuning iterations without overfitting and degeneration. The generated renditions of the desired subject preserve even fine details such as text and logos; all while maintaining the ability to generate diverse samples that follow the input text prompt. Since our method focuses on data augmentation, rather than adjusting the model architecture, it is complementary and can be combined with prior work. We show on established benchmarks that our data-centric approach forms the new state of the art in terms of image quality, with the best trade-off between identity preservation, diversity, and text alignment.

## 1 Introduction

Recent text-to-image diffusion models have made significant strides in creating realistic, diverse, and precise images from text inputs (Rombach et al., 2022; Saharia et al., 2022; Podell et al., 2023). However, their capability to accurately represent specific subjects, like unique backpack designs or individual dog breeds, is constrained by the textual descriptions provided. This limitation has spurred substantial interest and investigation in the research community, particularly in the quest to generate fresh and contextually adaptable images of the same subject matter.

To tackle this issue, previous research has explored several promising approaches. Text-conditioned image editing methods (Brooks et al., 2023; Kawar et al., 2023) enable image manipulation based on text input while retaining the subject's identity. However, these methods struggle when tasked with generating entirely new images in different contexts. On the other hand, image composition techniques (Wu et al., 2019; Lin et al., 2018; Cong et al., 2020; Yang et al., 2023) can transplant a given subject into alternative backgrounds but cannot create the subject within a novel scene from text alone. Recently, researchers have shown promise by fine-tuning large-scale diffusion models to associate an identifier token with the desired subject. This not only preserve subject identity but also exhibit impressive generalization across various textual inputs (Ruiz et al., 2023a; Gal et al., 2022). Nevertheless, these methods still face challenges in reproducing fine details of the subject and are susceptible to overfitting to the limited training data.

One straightforward method to enhance detail preservation is training longer. However, this raises the challenge of mitigating overfitting. Previous studies have suggested various strategies, such as fine-tuning specific model components (Hu et al., 2021; Kumari et al., 2023; Tewel et al., 2023; Qiu et al., 2023; Han et al., 2023; Voynov et al., 2023; Sohn et al., 2023). In contrast, our contribution is a data-centric approach that leaves the model architecture unchanged.

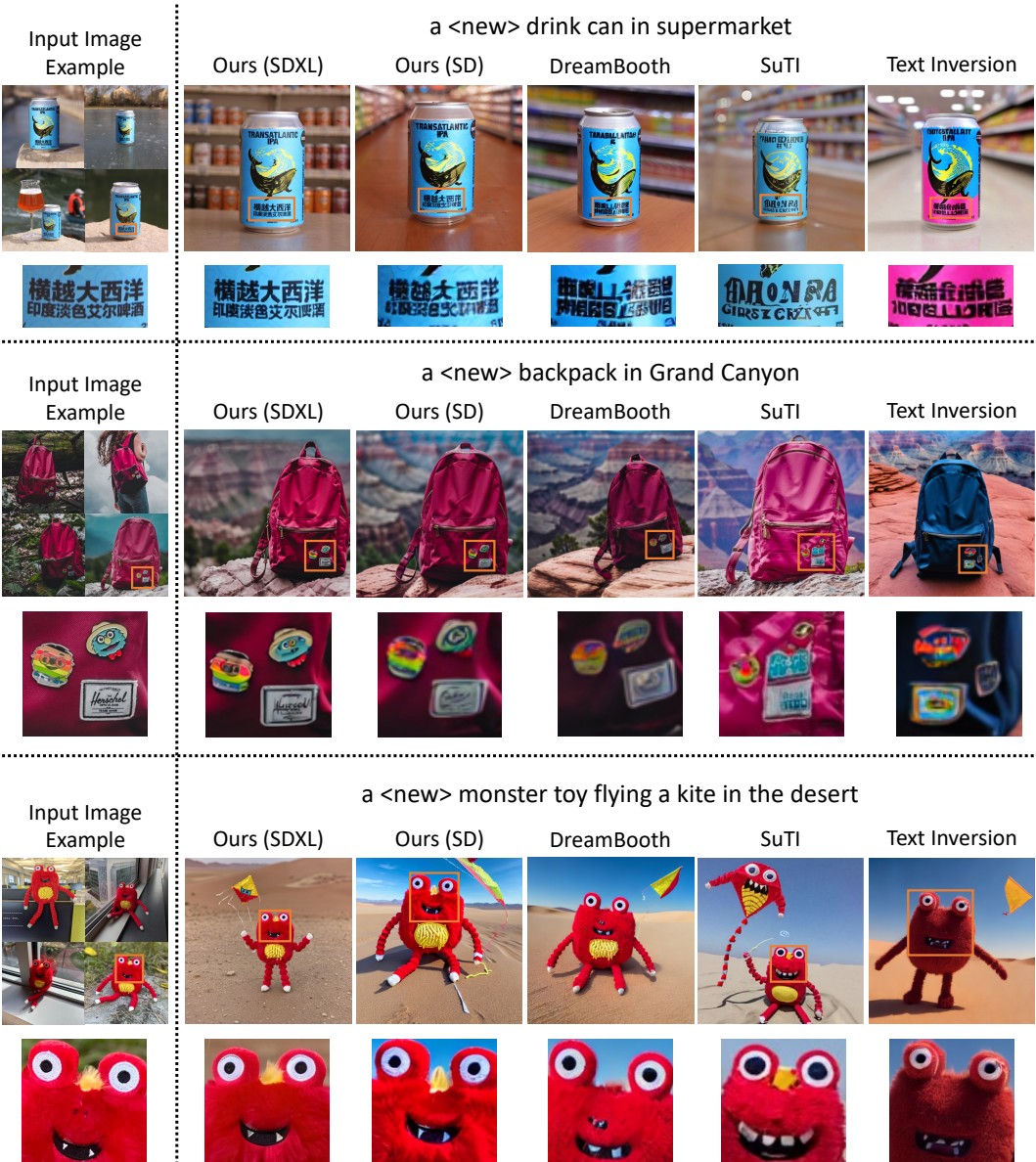

Figure 1: Demonstration of our method applied to both SDXL and SD, compared with several prior works. Note that our results preserves very fine details of the subject without overfitting to the training images and losing text alignment.

We introduce a novel technique for creating a regularization dataset tailored to counteract overfitting. Fundamental to our method is the observation that the fine-tuning not only associates the subject to the identifier token but also to the same simple sentence structure used for fine-tuning, limiting the ability to use more complex prompts at test time. To this end, we combine LLMs and text-to-image diffusion models to automatically generate prompts that are diverse in content and differ slightly in sentence structure to subsequently generate corresponding images that show the subject class in very diverse context. Using these synthetic examples for regularization yields a more stable fine-tuning process that can continue longer without overfitting and significantly enhances both identity preservation and text alignment. As demonstrated in Figure 1, our approach excels in preserving subject-specific details while maintaining robust generalization across diverse textual contexts.

This paper validates the effectiveness of our approach through extensive experimentation on the DreamBench dataset (Ruiz et al., 2023a), which contains a broad spectrum of subjects, including

complex items like drink cans and backpacks, as well as living animals with diverse poses and movements. Our experiments encompass a range of tasks, including subject re-contextualization, attribute modification, accessorization, and style transfer. Quantitative assessments using CLIP-I/CLIP-T and DINO scores, coupled with comparisons against DreamBooth, confirm our method's ability to proficiently preserve subject identity, including intricate details like logos, while demonstrating robust generalization across diverse textual inputs. Additionally, we highlight limitations in the original CLIP-T score used in DreamBooth Ruiz et al. (2023a) and propose an enhanced version of CLIP-T tailored to offer more precise and informative indications of text-image alignment.

We summarize our contributions in the following aspects:
1. A data-oriented approach to tackle overfitting issues during diffusion model fine-tuning.
2. A modification to the CLIP-T score to enhance its capacity for indicating text alignment.
3. A systematic exploration of diverse methods for generating a regularization dataset, with the intention of providing valuable insights to the broader research community.

## 2 RELATED WORK

**Diffusion Based Text-to-Image Models** have recently demonstrated remarkable progress, primarily through the utilization of diffusion models (Ho et al., 2020). These achievements are particularly pronounced when employing large models trained on extensive datasets (Rombach et al., 2022; Podell et al., 2023; Saharia et al., 2022; Balaji et al., 2022; Nichol et al., 2021; Ramesh et al., 2022). Our method extends the pre-trained diffusion models to incorporate personalized concepts. We apply this method to well-established models, such as StableDiffusion (Rombach et al., 2022) and StableDiffusionXL (Podell et al., 2023). Note that our method is fundamentally rooted in data, making it potentially applicable to a wide range of different diffusion model architectures.

**Text-to-Image Personalization** aims to imbue pre-trained diffusion models with the ability to produce novel images of specific subjects (Ruiz et al., 2023a; Gal et al., 2022). However, previous methods grappled with a delicate balancing act, oscillating between underfitting the intended subject and overfitting to the training dataset and losing text alignment. Attempting to address the problem of underfitting the subject, Hao et al. (2023) propose to inject a reference image feature map into attention modules to preserve identity. Chen et al. (2023a), on the other hand, embark on the task of disentangling subjects from backgrounds, employing a CLIP image encoder for encoding backgrounds (Radford et al., 2021). To combat the issue of overfitting, researchers have explored diverse strategies, fine-tuning different decomposition parts of the model (Hu et al., 2021; Kumari et al., 2023; Tewel et al., 2023; Qiu et al., 2023; Han et al., 2023; Voynov et al., 2023; Sohn et al., 2023). To sidestep time-consuming finetuning the diffusion model, several a encoder-based methods have emerged, involving the training of an additional encoder for all subjects (Shi et al., 2023; Gal et al., 2023; Ruiz et al., 2023b; Arar et al., 2023; Li et al., 2023a). These encoder-based methods often yield increased diversity in generated content, albeit with some trade-offs in identity preservation compared to fine-tuning-based approaches. Our approach is focuses on maximizing quality at the cost of training time, improving both identity preservation and text alignment relative to prior work. We leave the topic of accelerated training as an avenue for future exploration.

**Involving More Data** is our core idea. Previous work (Wang et al., 2023) has shown that a better prompt can significantly improve the generated image quality. The most similar approach to ours is proposed in DreamBooth Ruiz et al. (2023a). They propose generating a prior dataset using prompt "a [class noun]", e.g., "a backpack", to alleviate overfitting. However, our experiments reveal that this generated dataset lacks the strength to effectively counter overfitting. Kumari et al. (2023) extract the real text-image pairs that similar to the training examples from existing dataset, but they still need to restrict the number of training iterations to mitigate overfitting risks. Chen et al. (2023b) speed up customization with a feed forward network trained on many datasets of custom objects, prioritizing speed over quality improvements. Ma et al. (2023) propose to generate masks, bounding boxes, and their corresponding tags to train a better adapter and thus a better composition method. Our work is primarily centered on enhancing generation quality, with the aspect of speed left for future exploration.

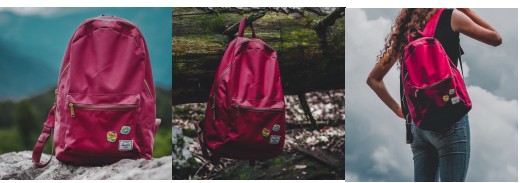 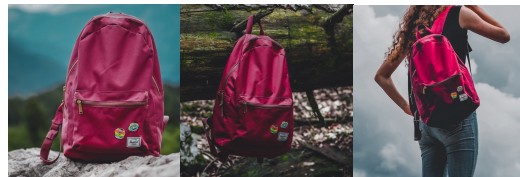

1. a <new> backpack
2. a <new> backpack
3. a <new> backpack

Original training examples

1. a <new> backpack on a rock
2. a <new> backpack hanging on a tree
3. a <new> backpack on a women's back

Step1: Add background prompts for example images

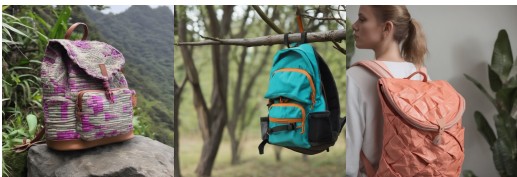 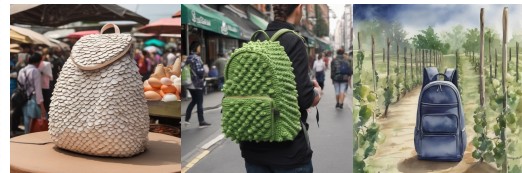

1. a contoured orchid woven backpack on a rock
2. a semicircular turquoise canvas backpack hanging on a tree branch
3. a rhomboid coral crinkled backpack on a women's back

Step2: Sample different subjects with the same class noun and same background

1. a cone-shaped eggshell scaly backpack amidst the aromas of a lively outdoor market
2. a spiky green knitted backpack in the heart of a bustling culinary street
...

Step3: Sample different subjects with the same class noun and different background

Figure 2: **Overview.** In contrast to prior approaches, we introduce specific prompts to the training examples and create a regularization dataset with a wider range of images guided by these prompts. To further boost diversity of regularization dataset, we generate additional prompts and images using the same prompt formats.

## 3 METHOD

Our main goal is to fine-tune pre-trained diffusion models using a limited set of subject-specific training examples, typically around 4 to 6 images. This fine-tuning process is aimed at enabling these models to generate new images of the same subject. However, preventing the model from overfitting to the training examples is a challenging task.

**Recap of DreamBooth**. Instead of linking the new subject with a noun word (Gal et al., 2022), Ruiz et al. (2023a) propose to linking the new subject with a adjective word, by inserting a special token before the subject noun. For example, "a backpack" becomes "a [special_token] backpack". Fine-tuning the diffusion models based on this newly designed prompt significantly improves the results. Furthermore, to prevent overfitting, they propose create a dataset of text-image pairs, employing prompts in the format "a [class noun]" (e.g., "a backpack"). This dataset serves as a counterbalance to the training examples. During training, both the training examples and the regularization examples are fed into the diffusion model. However, we discovered that the regularization effect is limited as the generated images and their prompts lack diversity.

**Our Approach**. In contrast to previous methods using a simplistic prompt for the real training examples, we first introduced individual and more concrete prompts, such as "a backpack on a rock". Surprisingly, this adjustment did not yield significant improvements, and the model still exhibited rapid overfitting. It became effective only when combined with a strong regularization dataset using prompts that describe foreground and background in more detail. Thus, our focus shifted towards automatically constructing an appropriate regularization dataset to address this form of overfitting. The overview of our method is presented in Figure 2. Importantly, our approach retains the model architecture while concentrating on the development of a novel regularization dataset to combat overfitting effectively. This means that it can be seamlessly combined with any personalization algorithm using a prior preservation dataset.

**Enhancing Training with Prompts**. The inclusion of prompts in training instances plays a pivotal role in our approach. While employing a uniform prompt format such as "a [identifier] [class noun]"

for all training examples may seem straightforward, it inadvertently increases the risk of overfitting to the training data. To address this challenge, we develop this idea further by incorporating background descriptions into each image, as depicted in the upper-right corner of Figure 2. Additionally, we replace the provided class name with more specific ones, which will be discussed further in Appendix B. It's important to note that merely adding the background prompt in isolation does not yield the desired benefits. Instead, our contribution is on constructing a dedicated regularization dataset, which proves to be essential in mitigating overfitting, as elaborated in subsequent sections.

**Generating against training prompts**. We present a novel approach for generating a regularization dataset, which builds upon the original example prompt. For instance, let's consider the initial prompt "a backpack on a rock". We enhance this prompt by introducing structural components in the format of "a [shape] [color] [texture] [class noun] [background]", where [shape], [color], and [texture] represent randomly selected adjectives drawn from a pool of 100 options for each attribute. These adjectives are then combined to create prompts for image generation, resulting in prompts such as "a contoured orchid woven backpack on a rock" and "a semicircular turquoise canvas backpack hanging on a tree branch", as demonstrated in the lower-left corner of Figure 2. The pools of the adjectives are generated automatically by Large Language Models (OpenAI, 2023). For more detailed information on this process, please refer to Appendix A.

**Amplifying Diversity with Structured Prompts**. To increase the diversity of our regularization dataset, we randomly generate 500 additional background phrases and 100 style phrases. We then combine these phrases in a random manner with the object-related prompts, resulting in structured prompts in the format "a [style] [shape] [color] [texture] [class noun] [background]". For instance, this approach generates prompts like "a photo of a cone-shaped eggshell scaly backpack amidst the aromas of a lively outdoor market" and "a children's storybook illustration of a trapezoidal coral embossed backpack against the canvas of a city skyline", as showcased in the lower-right corner of Figure 2. This strategy substantially enhances image diversity compared to the simplistic use of the prompt "a [class noun]".

**Adaption for Living Entities**. For living entities, we use descriptors like body, skin/fur, and emotion instead of shape, color, and texture. Additionally, we introduce motion into the backgrounds, transitioning from static scenes like "in an urban city" to dynamic contexts such as "walking in an urban city". For a detailed explanation of this adaptation process, please refer to Appendix A.

**Dropout**. In our strategy to enrich the dataset, we include three descriptive words for each object, which increases diversity. However, this approach can lead to a potential issue: overfitting to sentence structure. The model might learn that one adjective word corresponds to a specific subject, while three adjective words signify a different subject, impacting its ability to generalize. To mitigate this, we introduce randomness by randomly excluding some words during training. For example, we may have a prompt like "a [shape] [color] backpack on a rock" excluding a texture descriptor. This dropout technique adds an element of unpredictability, strengthening the model's adaptability and reducing its reliance on fixed sentence formats.

**Cropping**. Tewel et al. (2023) point out that the model is prone to overfit to the image layout when attempting to learn personalized concepts from a limited set of examples. Similarly, Kumari et al. (2023) observe that employing random cropping enhances convergence speed and yields improved results. To enhance our model's performance, we incorporate random cropping with a variable ratio, ranging from 0.75 to 1. It's worth noting that for SDXL, we incorporate cropping coordinates as described by Podell et al. (2023), where we increase the original image size to achieve a cropped image size of 1024x1024 pixels.

**Implementation Details** We create 2000 regularization images for each subject, distributed as follows: 20% from training prompts, 60% photorealistic images, and 20% styled images. We opt for the identifier word "olis" instead of the more commonly used "sks". This choice is based on the fact that "olis" corresponds to the least frequently utilized token in the model's vocabulary (2kpr, 2022). Each training batch contains one example from training set and one example from regularization set. For SD, we fine-tune the entire model with a learning rate of 2e-6 and perform inference using 200 steps of DDIM (Song et al., 2020). For SDXL, which has a larger model size, we employ a LoRA with a rank of 32 for both the text encoders and UNet. We also train the text embeddings. We set learning rate to 1e-4. We use 50 steps of DDIM for inference. While it's worth noting that different datasets exhibit varying degrees of diversity, we find that employing different early

stopping points can enhance diversity. For simplicity, we use 4000/8000 iterations for SD/SDXL in this paper. Additional details regarding the optimal iteration range can be found in Appendix F.

# 4 EXPERIMENTS

We employ the DreamBench dataset (Ruiz et al., 2023a), featuring 30 subjects ranging from backpacks and stuffed animals to dogs and cats. The dataset encompasses 25 distinct testing prompts for inanimate objects and 25 for living entities. Following the original paper, we generate four images per prompt, yielding a total of 3000 images for evaluation. Importantly, the testing prompts used in our experimental results are included in our regularization dataset.

Our baseline model is DreamBooth (SD backbone) (Ruiz et al., 2023a). For a fair comparison, we take the following steps: 1) Employ SDXL to generate 2000 high-quality regularization images; 2) Use "olis" as the identifier word; 3) Set the learning rate to 2e-6 for 1500 iterations; and 4) Apply 200 steps of DDIM for inference.

## 4.1 RESULT COMPARISON

The qualitative comparison is depicted in Figure 1. Our approach outperforms alternative methods in preserving intricate subject-specific details, ensuring identity retention. This is evident in the preservation of fine elements like logos on drink cans, unique icons on backpacks, and fine hair textures on toy monsters. While other methods have their strengths, they may struggle with preserving these details, emphasizing the superior subject fidelity of our approach.

| Methods | Backbone | DINO ↑ | CLIP-I ↑ | CLIP-T (vague class) ↑ | CLIP-T (object) ↑ |
|---|---|---|---|---|---|
| Real Image (on training set) | - | 0.774 | 0.885 | 0.298 | 0.311 |
| Textual Inversion Gal et al. (2022) | SD | 0.569 | 0.780 | 0.255 | - |
| Textual Inversion (our impl) | SD | 0.611 | 0.772 | 0.267 | 0.289 |
| DreamBooth Ruiz et al. (2023a) | SD | 0.668 | 0.803 | **0.305** | - |
| DreamBooth (our impl) | SD | 0.682 | 0.808 | 0.301 | 0.306 |
| Ours | SD | 0.704 | 0.824 | 0.293 | 0.309 |
| Ours | SDXL | **0.744** | **0.842** | 0.297 | **0.312** |

Table 1: Evaluation on the DreamBench.

| | Ours (SD) vs DreamBooth | Ours (SDXL) vs Ours (SD) |
|---|---|---|
| Subject Alignment | 64.9% / 35.1% | 67.8% / 32.2% |
| Textual Alignment | 52.8% / 47.2% | 46.8% / 53.2% |

Table 2: Human preference comparison

In our quantitative assessment, we employ DINO (Caron et al., 2021) and CLIP-I (Radford et al., 2021), following DreamBooth, to evaluate subject fidelity. For text alignment evaluation, we use CLIP-T, along with a modified version CLIP-T (object) to better estimate the text alignment. The details can be found in Appendix B.

We conduct a user study to compare with DreamBooth (Ruiz et al., 2023a). We asked 20 people to answer 30 questions about subject alignment and 30 questions about text alignment, in a total of 1200 questions. The questions are randomly sampled from a large pool of DreamBench test prompts. The details can be found in Appendix C. As shown in Table 2, our method is overwhelmingly preferred for subject alignment and comparable in text alignment. We also compared the effect of a different base model with SDXL instead of SD. SDXL is more expressive in details and hence leads to better subject alignment. For textural alignment SD works slightly better.

## 4.2 MODEL ANALYSIS

**How important is the formatted prompt?** We conducted an ablation test with two variations: 1) without using a regularization dataset, and 2) using a regularization dataset with simple prompts in

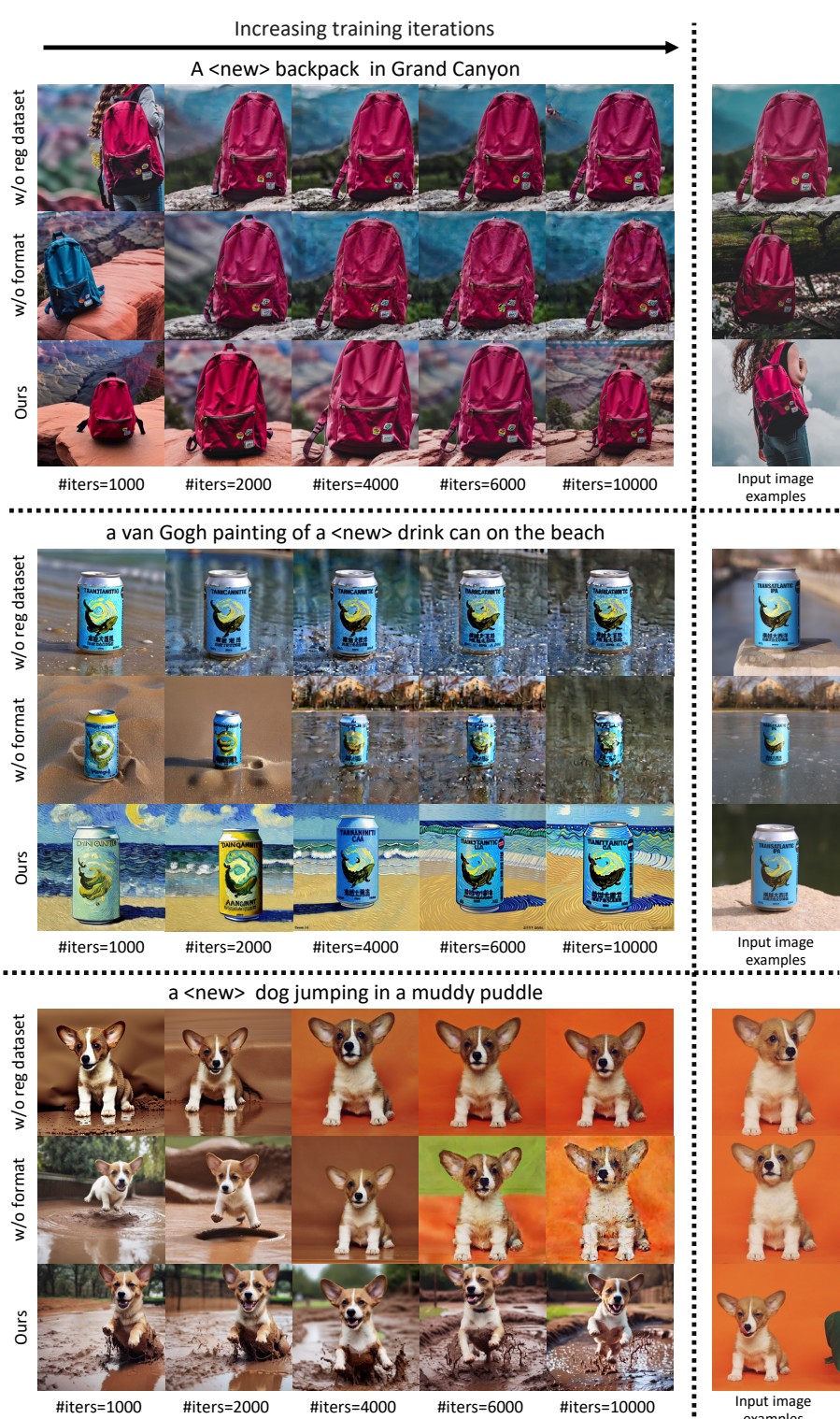

Figure 3: **Overfitting Prevention**. Our regularization dataset effectively prevents the model from overfitting to the training images. Interestingly, when we employ a regularization dataset with a simplistic prompt "a [class noun]", the model exhibits improved text alignment during initial iterations, but experiences a decline in performance over time.

the format "a [class noun]". The results are illustrated in Figure 3. These findings emphasize the importance of both the regularization dataset and the utilization of well-structured prompts. Notably, when employing a regularization dataset without well-structured prompts, there is an initial increase in diversity during early training phases compared to scenarios without any regularization dataset. However, over extended training iterations, a noticeable decline in model performance becomes evident. We attribute this decline to the incongruity between the overly simplistic prompt format "a [class noun]" and the complexities of the generated images. This observation underscores the pivotal role of the well-structured prompts in enhancing model performance, thus highlighting their significance in our approach.

**Does the model overfit to the formatted prompts?** To assess the impact of prompt phrasing variations, we rephrase the prompts and compare them to the originals, as illustrated in Figure 4. While our model consistently employs formatted prompts starting with "a <new>[class noun]" during training, it demonstrates resilience against overfitting to this specific format. In contrast, DreamBooth displays a notable departure from subject identity, confirming our assertion that the simplistic use of unvaried prompts in the format "a [class noun]" for constructing a regularization dataset can lead to overfitting to sentence structure. Additionally, it is noteworthy that SDXL exhibits a higher degree of consistency in generating images in response to rephrased prompts, with minimal deviation from the originals, thereby effectively preserving the underlying image structure.

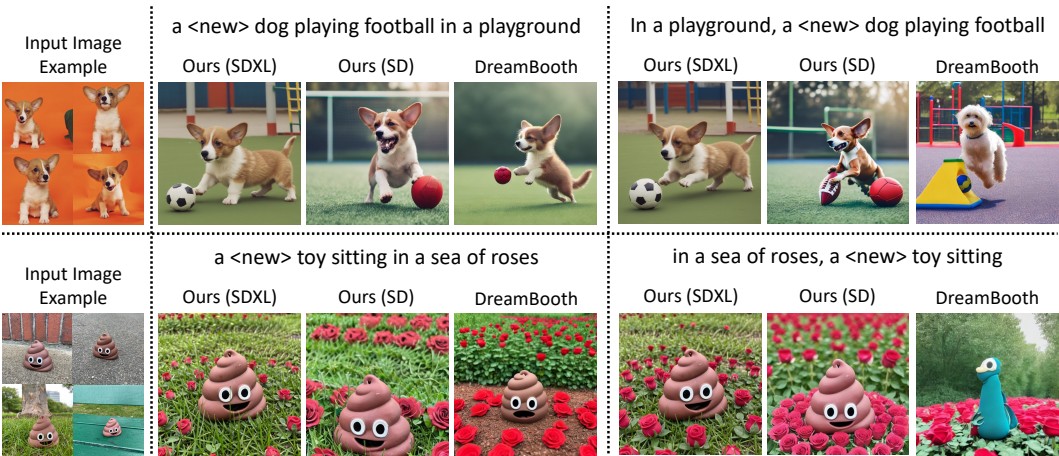

Figure 4: **Rephrasing prompts**. Even when using rephrased prompts, our method maintains subject identity, a quality DreamBooth lacks. Notably, SDXL consistently generates images in response to rephrased prompts with minimal deviation from the originals.

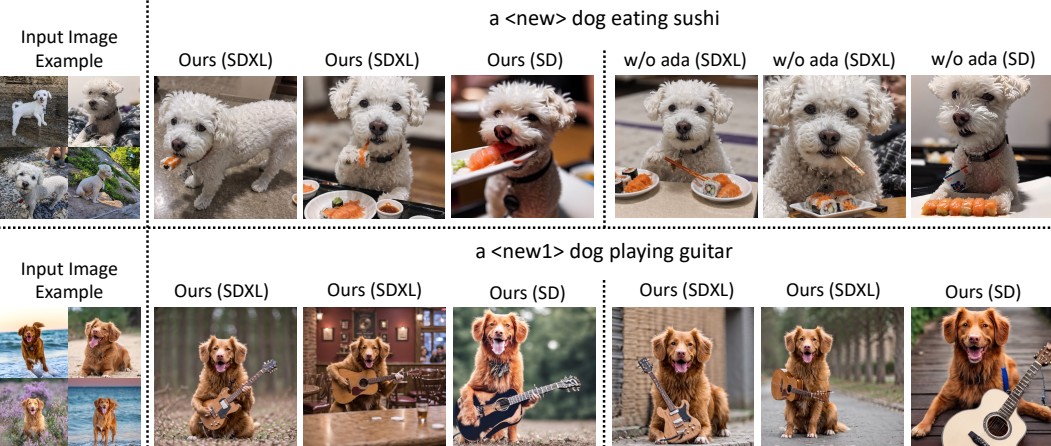

Figure 5: **Influence of adaption to living entities**. Without adaptation, the model might overlook motion in the prompts and focus solely on assembling objects within the images.

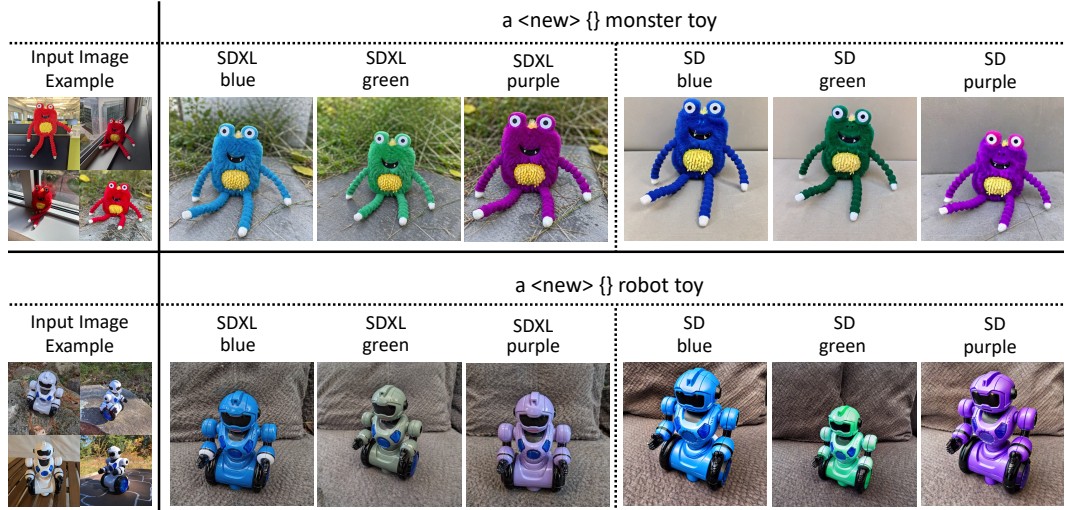

Figure 6: **Color Modification**. Our method can alter the color of the subject. It is important to mention that when modifying the color, using "a <new>[color] [class noun]" is more effective than "a [color] <new>[class noun]".

**Does the adaption to living entities help?** The key difference between inanimate objects and living things is motion. Training without adaptation tends to overlook motion while still generating objects separately, as shown in Figure 5. However, it's important to note that adaptation makes generating motion easier, but the model can still produce motion even without it.

**Is the random dropout of adjective words necessary?** When altering attributes like color, we found that using random dropout is essential. Without it, the model tends to excessively preserve the original identity, making color changes challenging. With random dropout, the task becomes easier while still preserving identity, as shown in Figure 6. It's worth noting that for color modification, using "a <new>[color] [class noun]" works better than "a [color] <new>[class noun]".

## 5    LIMITATIONS

Despite its impressive quality, our method has two limitations: 1) It needs to generate a regularization dataset for each category, which increases the overall training time (in practice <500 images also work for simple cases); 2) On a single A100 GPU, it requires approximately 1.5 hours for SD and 3.5 hours for SDXL to complete 4000 and 8000 training iterations, respectively. We leave acceleration to future research. Currently, we assume that the real images of the target object are annotated manually. In the future, caption generator could be used, such as BLIP (Li et al., 2022; 2023b).

## 6    CONCLUSION

We introduced a data-centric approach to enhance diffusion model personalization, using a structured regularization dataset. By incorporating formatted prompts and their generated images, this method effectively reduces overfitting, allowing for up to a 5x longer training duration. This results in significant improvements in image quality, identity preservation, and diversity of generated samples aligned with input text prompts. Although this approach requires extended training times and increased memory resources, we foresee potential solutions in the future, as our method sets the stage for forthcoming data-centric approaches. Notably, our method complements existing work by focusing on data augmentation rather than model architecture adjustments, making it potentially compatible for integration with previous research. For future work, we would like to focus on 1) generalize to multi-concepts; 2) apply on videos; and 3) accelerate the procedure.

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

Figure 7: **CLIP-T score on different subject names**. We randomly generate an image with the prompt "a teddy bear toy sitting beside a river". Subsequently, we evaluate prompts "a  in the grass next to a tree" and "a  sitting on a sofa". When employing the vague prompt "toy", both the ground truth and the mismatched example are classified as mismatches (<0.3). In contrast, when using specific class names, the ground truth is categorized as a match (>0.3), while the mismatched examples remain classified as mismatches.

## A    FORMATTED PROMPT GENERATION

For non-live objects, we use the following prompts to generate words describing shape, color, textures, and background in ChatGPT:

- shape: give me 100 adjective words describing the shape of an object
- color: give me 100 adjective words describing the color of an object
- texture: give me 100 adjective words describing the texture of an object
- background: give me 500 phrases that describe the background, such as "on the table", as diverse as possible.

After removing duplicated ones, there are 85 shapes, 93 colors, 96 textures, and 455 backgrounds.

For live objects, we use the following prompts to generate words describing shape, color, textures, and motion in ChatGPT:

- body: give me 100 adjective words describing the body of an animal
- skin: give me 100 adjective words describing the skin or fur of an animal
- emotion: give me 100 adjective words describing the emotion of an animal
- motion: give me 1000 different short concise sentences that contains a special token "$concept" which stands for a specific animal, which can be a dog, a cat or a human. For example: "a $concept sitting in a temple", "a $concept walking in a supermarket". Keep "a $concept" in the sentences.

After removing duplicated ones, there are 89 bodies, 86 skins/furs, 75 emotions, and 744 motions. For humans, we replace the word "animal" above with "person".

We use

- style: give me 100 image style descriptions, such as "a photo of", and "a painting of".

After removing duplicated ones, there are 99 styles left.

## B    BETTER CATEGORY NAMING

We argue that the CLIP-T used in DreamBooth may yield inaccurate measurements of text-image alignment when vague class names are employed. As shown in Figure 7, there is a notable discrepancy between the utilization of vague class names, such as "toy", and more specific object names, such as "duck toy", on the ground truth images. Notably, the CLIP-T score appears to be significantly influenced by the nomenclature chosen for the object, thereby potentially undermining its

accuracy as an indicator of text-image alignment. To delve deeper into this matter, we calculate the CLIP-T score on the original images and the manually added prompts. Table 1 presents that, when using vague class names, the CLIP-T score for ground truth text-image pairs falls even below the conventional threshold of 0.3, typically considered as the threshold for assessing text-image pair compatibility Schuhmann et al. (2021). To rectify this issue, we replace vague class names with highly specific object names, resulting in a substantial improvement in the CLIP-T score for ground truth text-image pairs.

We show the name change in Table 3. As shown in Section 4.1, the CLIP-T score makes more sense in the changed name than the original name.

| subject name | original class | modified class |
|---|---|---|
| bear_plushie | stuffed animal | bear plushie |
| berry_bowl | bowl | berry bowl |
| can | can | drink can |
| clock | clock | alarm clock |
| duck_toy | toy | duck toy |
| grey_sloth_plushie | stuffed animal | sloth plushie |
| monster_toy | toy | monster toy |
| poop_emoji | toy | poop emoji toy |
| rc_car | toy | racing car toy |
| red_cartoon | cartoon | 2d cartoon devil |
| robot_toy | toy | robot toy |
| wolf_plushie | stuffed animal | wolf plushie |

Table 3: **Name Change**. We change the name for a more reasonable CLIP-T metric and better performance.

## C    DETAILS OF USER STUDY

We randomly sampled and paired 300 comparisons of ours(SD) versus DreamBooth, half of which is for the subject alignment and the other half for the text alignment. For subject alignment, we randomly sampled a ground truth image and asked "The foreground object in which image is more similar to the reference?". For text alignment, we asked "Which image better depicts {}?", where {} is replaced by the prompt. We equally divided the questions into 10 groups. Each person randomly received one group. We did the same for ours(SDXL) versus ours(SD).

## D    ABLATION TESTS ON TRAINING AND REGULARIZATION DATASET SIZE

We did grid experiments on number of training examples 1, 2, 4, and number of regularization examples 100, 500, 1000, 2000. The results are illustrated in Figure 8. Our method is robust to smaller regularization set size, as only 100 regularization examples can also effectively prevent overfitting and the model is still able to preserve the very fine details of the subjects.

Interestingly, when the subject is complex, only one or two training examples with a large amount of regularization examples ($\geq$500) may result in underfitting, such as the backpack in Figure 8 bottom. This is not observed when the subject is simple, such as the dog in Figure 8 top.

## E    EXTENSION: USING BLIP TO GENERATE CAPTIONS

We tried to use BLIP (Li et al., 2022) to generate more personalized captions for the training example images. BLIP outputs a caption for the input image and can be conditioned on the format. We condition BLIP so that it generates prompts that start with "a [subject]". For instance for the "tortoise plushie" image BLIP generates

- a tortoise plushie on a pillow
- a tortoise plushie

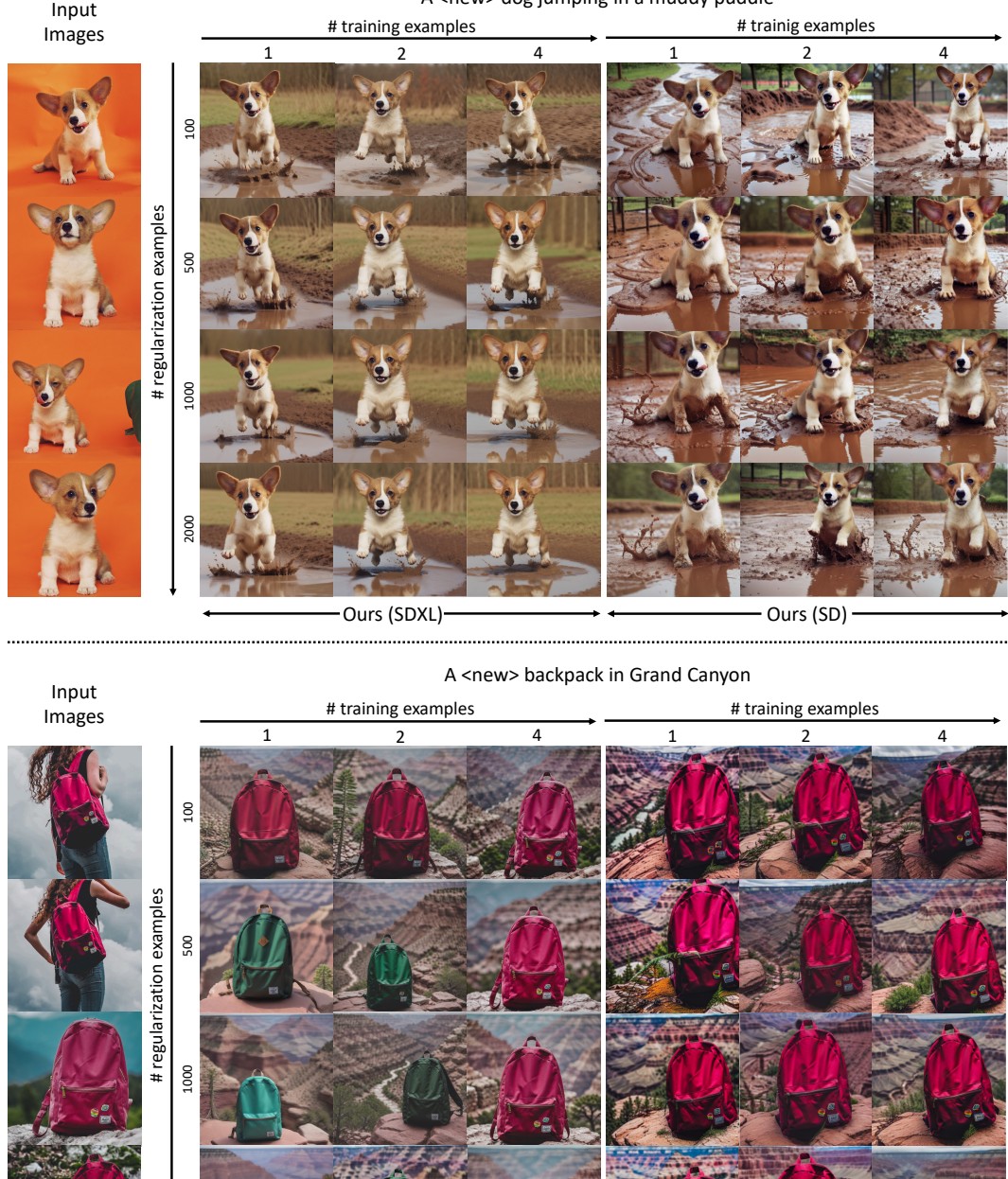

Figure 8: Ablation Tests on number of training examples and regularization dataset size.

- a tortoise plushie sitting on a piano keyboard

- a tortoise plushie on a desk

- ...

To unify the prompt format, we task ChatGPT to *'Change the following sentence to the format "A <new> tortoise plushie blablabla". The "<new>" is a special token that needs to be inserted before tortoise plushie.'* The result are the following prompts

- a <new> tortoise plushie on a pillow
- a <new> tortoise plushie
- a <new> tortoise plushie sitting on a piano keyboard
- a <new> tortoise plushie on a desk
- ...

The results of this "tortoise plushie" dataset is shown in Figure 14. With this addition of using BLIP, it alleviated writing the prompt examples manually, i.e., it replaced the manual steps in Section 3.

## F  OPTICAL NUMBER OF ITERATIONS OF DREAMBENCH

We show the name change in Table 4. As shown in Section 4.1, the CLIP-T score makes more sense in the changed name than the original name.

| subject name | best #iterations on SD | best #iterations on SDXL |
|---|---|---|
| backpack | 6000-8000 | 8000-10000 |
| backpack_dog | 2000-3000 | 4000-6000 |
| bear_plushie | 2000-4000 | 4000-6000 |
| berry_bowl | 6000-8000 | 8000-10000 |
| can | 6000-8000 | 8000-10000 |
| candle | 4000-6000 | 8000-10000 |
| cat | 1000-3000 | 1000-3000 |
| cat2 | 6000-8000 | 8000-10000 |
| clock | 6000-8000 | 8000-10000 |
| colorful_sneaker | 4000-6000 | 6000-8000 |
| dog | 1000-3000 | 1000-3000 |
| dog2 | 2000-4000 | 4000-6000 |
| dog3 | 2000-4000 | 8000-10000 |
| dog5 | 3000-4000 | 6000-8000 |
| dog6 | 3000-4000 | 6000-8000 |
| dog7 | 3000-4000 | 6000-8000 |
| dog8 | 1000-3000 | 1000-3000 |
| duck_toy | 3000-4000 | 3000-4000 |
| fancy_boot | 3000-4000 | 6000-8000 |
| grey_sloth_plushie | 3000-4000 | 6000-8000 |
| monster_toy | 3000-4000 | 8000-10000 |
| pink_sunglasses | 3000-4000 | 4000-6000 |
| poop_emoji | 3000-4000 | 4000-6000 |
| rc_car | 3000-4000 | 4000-6000 |
| red_cartoon | 6000-8000 | 8000-10000 |
| robot_toy | 3000-4000 | 6000-8000 |
| shiny_sneaker | 3000-4000 | 6000-8000 |
| teapot | 6000-8000 | 8000-10000 |
| vase | 6000-8000 | 8000-10000 |
| wolf_plushie | 3000-4000 | 4000-6000 |

Table 4: **Best #iterations of datasets in DreamBench.** The variation mainly comes from the diversity of the dataset itself.

## G  MORE RESULTS

Training Image Examples

a <new> {} in the snow

a <new> {} with the Eiffel Tower in the background

a <new> {} floating on top of water

Figure 9: **More Results on inanimate objects**.

Training Image Examples

a <new> {} on top of a wooden floor

a <new> {} with a city in the background

a <new> {} on top of a purple rug in a forest

Figure 10: **More Results on inanimate objects**.

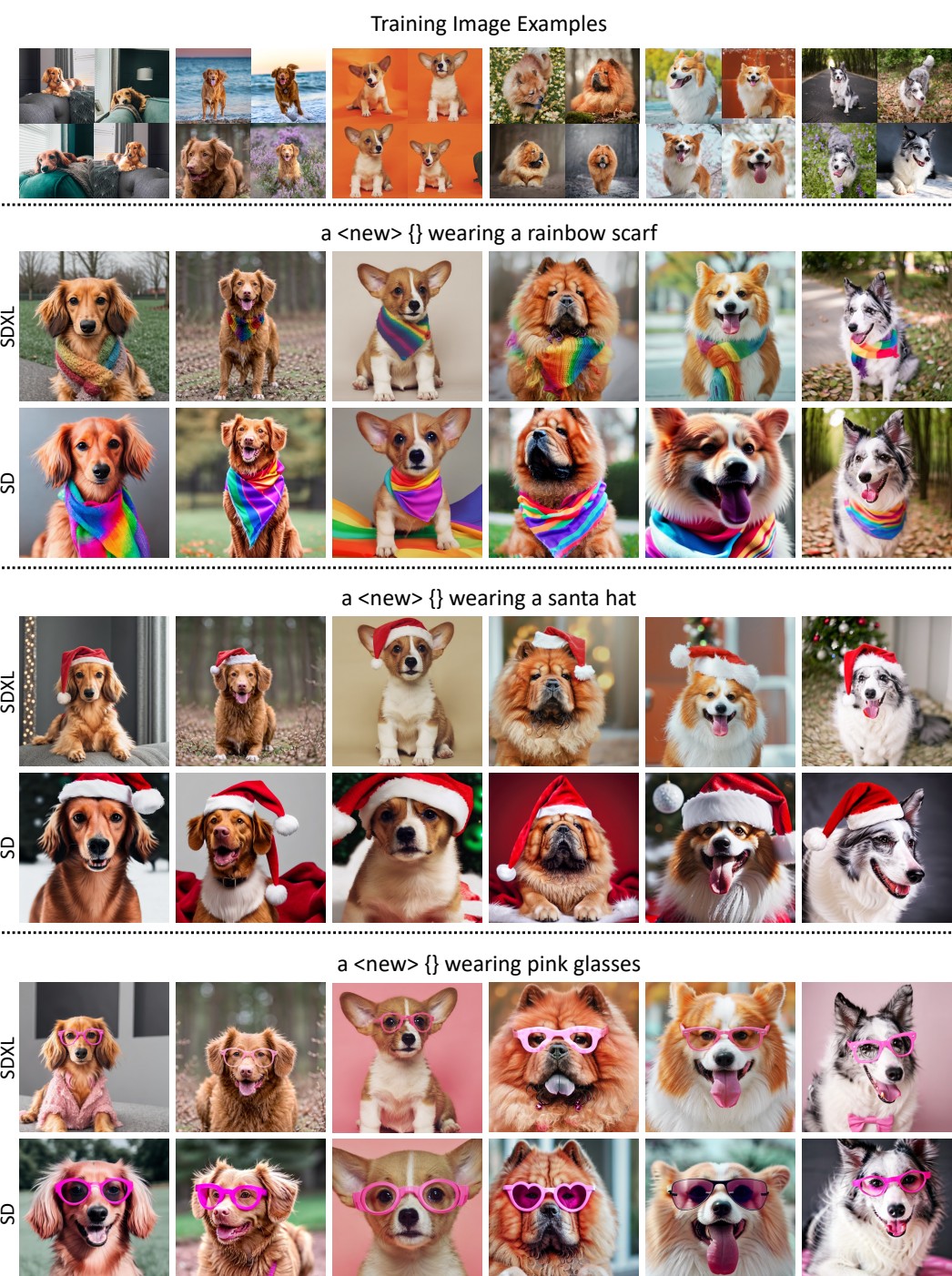

Figure 11: **More Results on living entities**.

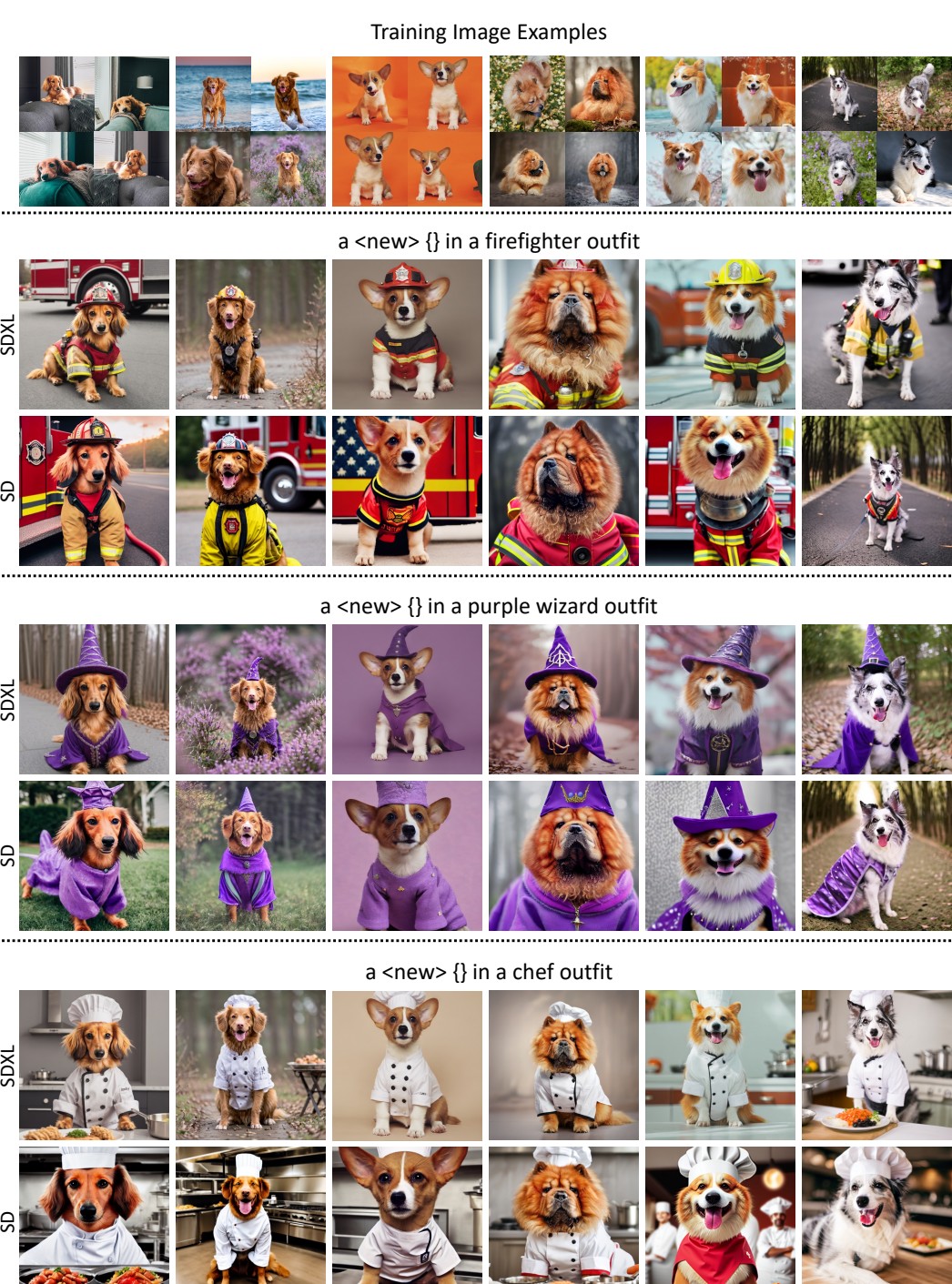

Figure 12: **More Results on living entities**.

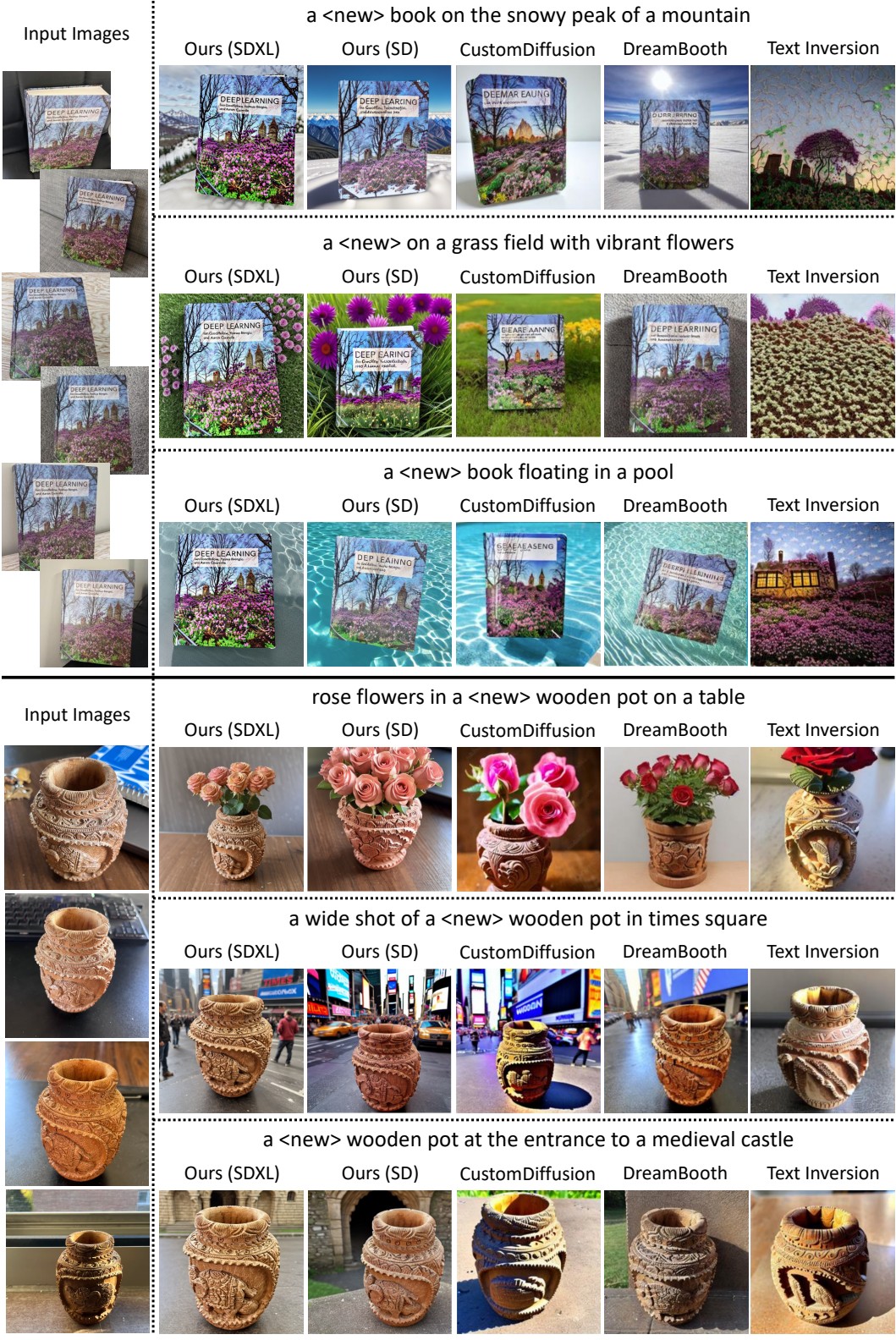

Figure 13: More Comparison.

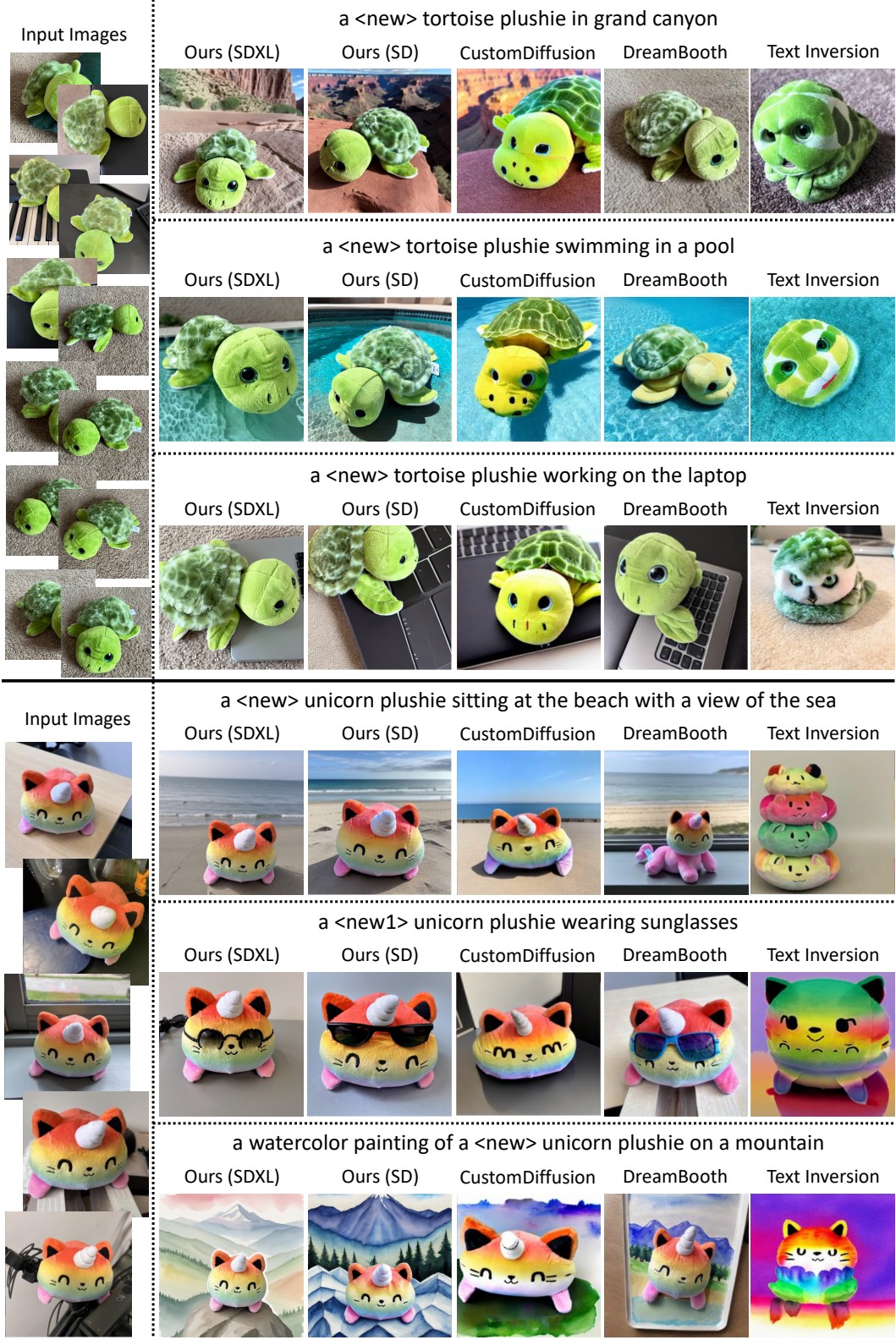

Figure 14: More Comparison.

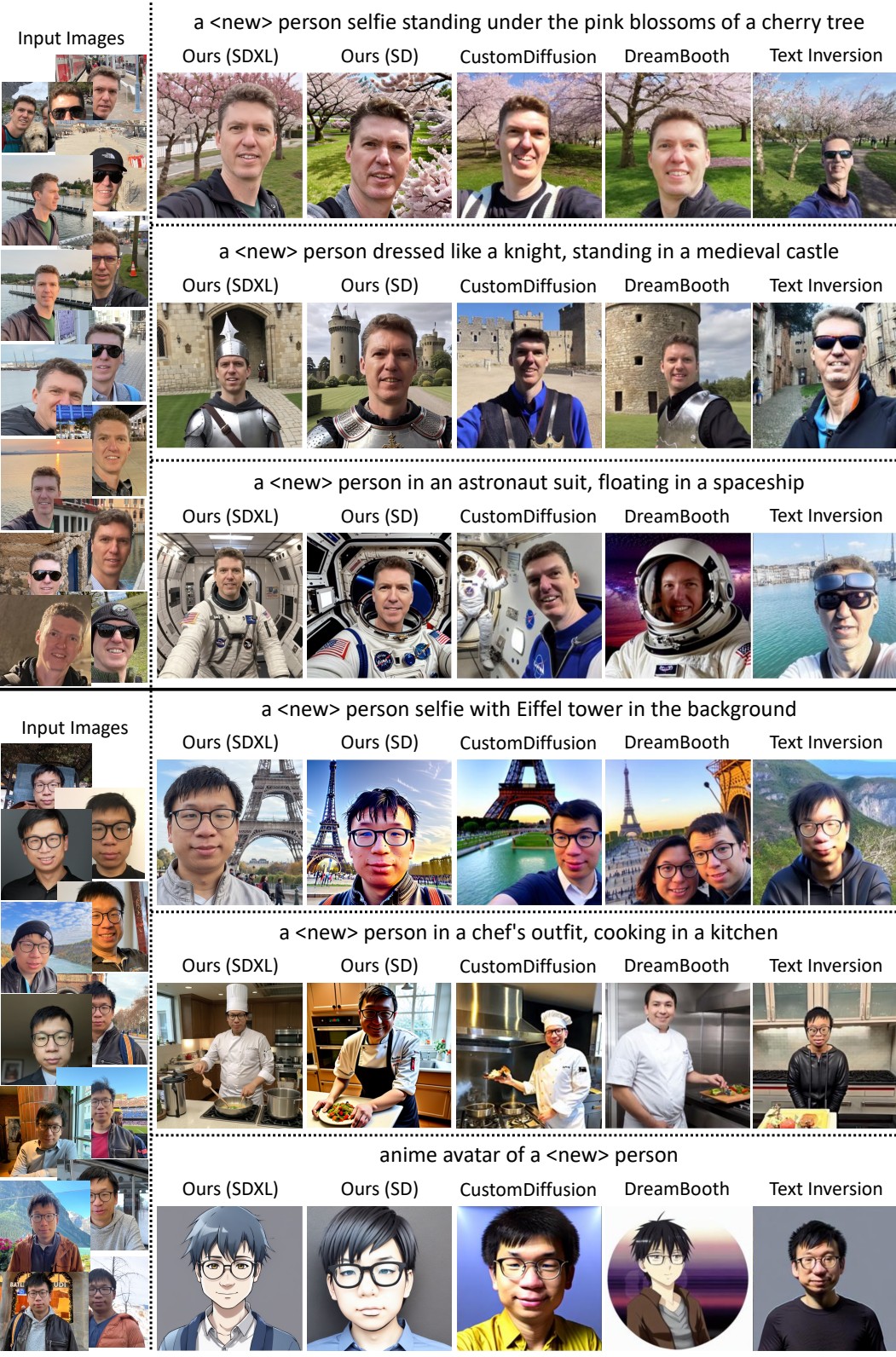

Figure 15: More Comparison.

