# OpenReview forum: "A Data Perspective on Enhanced Identity Preservation for Diffusion Personalization"
_ICLR.cc/2024/Conference — Submitted to ICLR 2024_

### Official Review · Reviewer_AXFR · 2023-10-28

**Soundness:** 3 good
**Presentation:** 3 good
**Contribution:** 3 good
**Rating:** 6
**Confidence:** 3

**Summary:**

This paper proposes a new method to inject new visual concepts into the generation, using few images. The authors propose a novel regularization dataset generation strategy on both the text and image level. The formulated dataset can help to prevent losing text coherence and prompt better identity preservation. The results are established on benchmarks, demonstrating the effects of the proposed method.

**Strengths:**

The proposed data approach is effective on the chosen benchmarks.

**Weaknesses:**

1.	The formatted prompt generation is limited to several categories. For example, according to the supp., for live objects, the prompts are all obtained via the subject of animal. How about the prompts for human? I do not think this prompt generation strategy is general enough.

2.	Moreover, I think the prompts should be generated according to different input images, employing multi-modality models.

3.	I think the new objects to be inserted into the generation in this paper’s experiments are few. More cases are needed to analyze the effects of the proposed method.

4.	In Fig.4, why the performance of w/o format will lead to worse results as the increase of iteration number? Even without the use of formatted prompts, the generation results should be more fitted with the target object along with the training.

**Questions:**

1.	I wonder the performance of the proposed method if there are fewer input examples, like 1-3 examples.

2.	Can the prompts be generated online with the training? It will save a lot of time.

3.	Few examples can not reflect the true quality of the generation, is there any subjective evaluation?

**Details Of Ethics Concerns:**

The generated images may be harmful to the protection of copyright.

---

> ### Author Response · Authors · 2023-11-23
> **Official Rebuttal**
>
> Thank you for the helpful comments. We address each question below.
>
> ## W1: Generalization to other categories?
> We use object/animal as categories, which covers most kinds of living and non-living objects. For a particular application domain, one could always generate more specific prompts with minor modifications. It would only take a few seconds by using our LLM strategy. We add an example on humans and show the results in Figure 15 (updated manuscript) along with the comparison with other methods. It demonstrates that the same prompt generation works for humans and that our approach preserves better identity than other methods while still aligning with the text prompts.
>
> ## W2: Generating prompts from images instead of generating images from prompts?
> We agree that it is an interesting alternative direction, but it is dissimilar from our goals. One of our goals is to exploit the broad distributions over words and their relation learned by LLMs to make the image generation more diverse.
>
> ## W2 & Q2: Can the prompts be generated online with the training, from the training images, using multi-modality models?
> This is a promising direction that however bears its own challenges. We did an initial experiment on the “​​tortoise plushie” in Figure 14 (updated manuscript). We use the multi-modal model BLIP to generate the captions and then use ChatGPT to reformulate the sentence. Subsequently, we use an LLM to convert these into the correct format with the special <new> token inserted. We show the details in Appendix E (updated manuscript). This alleviates writing the prompt examples manually, i.e., it replaces the manual steps in Section 3. As shown in Figure 14, the results are as good as the ones with the manually created prompts and outperform other methods.
>
>
> ## W3: Need more qualitative results?
> We already have a range of qualitative results in Figure 9-12 (numbers referring to the updated manuscript). We added some more results for comparison in Figure 13-15. In addition, we evaluated on the DreamBench testsuite. We added this comparison to the supplementary materials. Due to the space limit (<100M), we only sample each prompt once for each subject, resulting in 750 comparison images.
>
> ## W4: Why does the experiment of “w/o format” lead to worse results as the increase of iteration number?
> It is a consistent finding as we also observed this before and analyzed in Sec 4.2 first paragraph “How important is the formatted prompt? …”. We additionally noticed that in the first 2,000 iterations, the simplistic regularization set with prompt “a [subject]” effectively prevents overfitting. However, it fails to do so with prolonged training and thus cannot preserve the fine details of the subjects. We believe it is due to the deviation between the simplistic prompt “a [subject]” and the complexity of the images. Such a simplistic prompt is not detailed enough to explain the image. Therefore, with longer training, the model degenerates. Interestingly, even without the regularization set, if the prompts of the training example are accurate instead of simplistic, the model overfits in terms of limited diversity but without degeneration, as shown in Figure 4 (w/o reg dataset).
>
> ## Q1: Performance of the proposed method with fewer input examples?
> We added the ablation tests, see common question #1.
>
> ## Q2:
> See W2 above.
>
> ## Q3: Missing human evaluation?
> We added one, see common question #3.

---

### Official Review · Reviewer_rHiC · 2023-10-31

**Soundness:** 3 good
**Presentation:** 3 good
**Contribution:** 3 good
**Rating:** 6
**Confidence:** 4

**Summary:**

This paper proposes a data-driven approach to improve personalized generation. The paper first discovers that previous class regularization is ineffective in alleviating overfitting due to a lack of diversity. The paper proposes to first enhance the training prompts associated with the concept images by including more specific class names and background descriptions. Based on the training prompts, the regularization prompts are further enhanced by introducing structural components including shape, color, and texture, which are then amplified with more diverse backgrounds and styles.

**Strengths:**

(1) The paper attempts to improve personalized T2I generation from a data-centric perspective, which focuses on automatically generating a rich and informative regularization dataset. Despite there exist a few methods that improve T2I generation via better prompting [1, 2] (note that [1] was submitted to arXiv on 25 Oct 2023, hence not in the scope of this work), this paper is the first work to improve personalized generation by diversifying the regularization data.

(2) This paper provides insights into the importance of the quality of the regularization dataset in order to prevent overfitting, and is complementary to existing works that attempt to improve architectures and training schemes. This may inspire future research on further improving diffusion personalization.

(3) The proposed method demonstrates a notable improvement in generating personalized images with higher fidelity and is capable of preventing overfitting especially when facilitating larger training iterations.


[1] Segalis, Eyal, et al. "A Picture is Worth a Thousand Words: Principled Recaptioning Improves Image Generation." arXiv preprint arXiv:2310.16656 (2023).
[2] Wang, Yunlong, Shuyuan Shen, and Brian Y. Lim. "RePrompt: Automatic Prompt Editing to Refine AI-Generative Art Towards Precise Expressions." Proceedings of the 2023 CHI Conference on Human Factors in Computing Systems. 2023.

**Weaknesses:**

(1) The baselines that are compared in this paper are textual inversion and DreamBooth, which are both pioneering works in diffusion personalization. However, there exist many more improved personalization methods, e.g. Custom Diffusion [3], that are also widely used. Experimenting based on more methods will further emphasize the generalizability and complementarity of the proposed method.

(2) On top of Custom Diffusion [3], it would be also interesting to see whether the data-driven approach can benefit multi-concept learning.

(3) The method requires generating a relatively large regularization dataset (containing ~2000 images), which inevitably leads to much longer training time.

Small TYPO:
1. TYPO in section 5, the first bullet in the first paragraph should be “1)” instead of “2)”

[3] Kumari, Nupur, et al. "Multi-concept customization of text-to-image diffusion." Proceedings of the IEEE/CVF Conference on Computer Vision and Pattern Recognition. 2023.

**Questions:**

Please see the weaknesses above.

---

> ### Author Response · Authors · 2023-11-23
> **Official Rebuttal**
>
> Thank you for the helpful comments and appreciation of our method. We address each question below.
>
> ## W1: New baseline CustomDiffusion?
> We added the qualitative comparison in Figure 13-15 (updated manuscript). Our generated images have higher subject alignment than theirs while having better or equal text alignment. According to our user study in comparison to DreamBooth, our method is preferred in subject alignment by 64.9%. An indirect comparison is possible to CustomDiffusion which, in an equivalent user study, scored only 56.62% compared to DeamBooth.  Due to the time limit, we will add direct comparison with CustomDiffusion in the camera-ready version.
>
> ## W2: Data-driven approach for multi-concept learning?
> It is indeed an interesting topic we would like to explore in the future. For this paper, we focus on the detailed subject alignment, i.e., identity preservation. We added this to our conclusion.
>
> ## W3: Too large regularization set?
> We kindly refer you to the question #1 in the common response. A smaller regularization set (<500) also works.
>
> ## Typo:
> Fixed, thx.
>
> ## Related Work
> Added and discussed in Section 2. Thx.

---

### Official Review · Reviewer_hHDA · 2023-11-01

**Soundness:** 2 fair
**Presentation:** 1 poor
**Contribution:** 2 fair
**Rating:** 3
**Confidence:** 3

**Summary:**

It seems (please refer to Weaknesses for reasons why I use the word "seem") that the authors introduced an extension of DreamBooth, which can better preserves the details of the objects of interest. The authors proposed to achieve this goal by generating a larger set of regularization images. Specifically, they seem to be generated using prompts that are (1) generated by a large language model (LLM) or (2) generated following specific templates. Experiments are conducted on DreamBench and metrics show that the proposed method generates higher quality images than existing methods.

**Strengths:**

1. Extensive visual comparison with existing methods are presented in the paper. It is evident that, for the examples provided, the proposed method better preserves the details of the objects of interest

2. The use of English is satisfactory.

3. Ablations are conducted to help readers understand several design choices made by the authors.

**Weaknesses:**

1. Poor presentation: I find this paper hard to follow. Several important aspects of the proposed method remain a mystery, e.g., how the 2000 "regularization images" are created, why the 2000 images are called regularization images (I am not able to see how they can regularize the model from "Each training batch contains one example from training set and one example from regularization set."). It seems to me that this paper tries to extend DreamBooth [Ruiz, 2023a] and the 2000 regularization images may be generated using Stable Diffusion and the 2000 samples may be used to compute the class-specific prior preservation loss to regularize the model. However, a reader needs to be very familar with DreamBooth in order to make these guesses and they are just guesses.

2. To me, this is a trival extension of DreamBooth. The authors proposed to generate more "regularization images" using prompts that are (1) generated by a large language model (LLM) or (2) generated following specific templates. It is hard for me to agree that this paper meets the bar for an ICLR paper.

3. Lack of human evaluation.

**Questions:**

1. How are the 2000 regularization images created? Are they generated by a pre-trained diffusion model? If so, why do you use the word "created"?

2. Why the 2000 images are called "regularization images"? How can they help regularize the model? If you follow DreamBooth [Ruiz, 2023a], please specifically mention this.

3. Would be great to see a comparison with DreamBooth + LoRA and Textual Inversion + LoRA.

4. How does the performance of the proposed method change when more number of samples are available, e.g., 20 samples? How does the performance of the purposed method compare with other methods when more number of samples are available?

---

> ### Author Response · Authors · 2023-11-23
> **Official Rebuttal**
>
> Thank you for the helpful comments. We address each question below.
>
> ## W1: Hard to follow without knowing DreamBooth?
> We sincerely thank you for pointing out the points where writing can be improved. To make it easier for readers, we created a separate subsection at the beginning of our method section (Section 3), to introduce DreamBooth in the updated manuscript.
>
> ## W2: Trivial extension to DreamBooth?
> DreamBooth also generates a dataset, which is called a prior set. Their purpose is to preserve the ability to generate similar subjects so that the model does not overfit. While they can train 2000 iterations without overfitting, the model still overfits and gets degenerated with longer training, as shown in Figure 4. Our insight is that their simplistic prompt “a [subject]” and its generated images are not powerful enough to prevent overfitting. Furthermore, the simplistic prompt “a [subject]” may not be enough to explain the diverse generated images. The deviation between the prompts and the images may result in model degeneration with prolonged training, as shown in Figure 4 (w/o format). Interestingly, even without the regularization set, if the prompts of the training example are accurate instead of being simplistic, the model only overfits without breaking down entirely, as shown in Figure 4 (w/o reg dataset). Our paper focuses on the importance of accurate prompts and their similar prompts while DreamBooth focuses on the importance of images and their similar images. Our prompts prevent the model from degeneration. They work as hard examples, forcing the model to relate the details with the special token. Instead of being a trivial extension to DreamBooth, our approach provides a new perspective to the problem, pushing the direction taken by DreamBoth further and deviating from the current trend to hand-tune model architectures. With the significant improvement we show, we expect a range of future work to follow in this underexplored direction.
>
> ## W3: Missing human evaluation?
> We added one, see common question #3.
>
> ## Q1 & Q2: Missing description of the regularization set generation process?
> While our main approach is stated in Section 3, the details of how to generate the prompts and the corresponding images are in supplementary A. To make it easier to find, we now concatenated it with the original paper. As stated in Section 4, the 2000 regularization images are generated from Stable Diffusion XL with structured but diverse prompts. We use the word “create” because the regularization set contains not only the images but also the created prompts.
>
> ## Q3: Comparison with DreamBooth + LoRA and Textual Inversion + LoRA?
> Investigating different model architectures and fine-tuning strategies is orthogonal to our goal of improving the data. To show that our method outperforms also the more recently developed models, we compare our method with the recent CustomDiffusion model. The results are illustrated in Figure 13-15 (updated manuscript). Our generated images have higher subject alignment than theirs while having better or equal text alignment. According to our user study in comparison to DreamBooth, our method is preferred in subject alignment by 64.9%. An indirect human comparison is possible to CustomDiffusion which, in an equivalent user study, scored only 56.62% compared to DeamBooth. Due to the time limit, we will add direct comparison with CustomDiffusion in the camera-ready version.
>
> ## Q4: Performance when more examples are available?
> We show in Figure 14-15, examples using theCustomConcept101(https://github.com/adobe-research/custom-diffusion/tree/main/customconcept101) dataset. With more than 10 examples, our method still outperforms other methods, especially in the subject alignment.

---

### Official Review · Reviewer_vEAP · 2023-11-05

**Soundness:** 2 fair
**Presentation:** 2 fair
**Contribution:** 1 poor
**Rating:** 3
**Confidence:** 4

**Summary:**

In this paper, the authors propose to perform prior preservation in personalized text-to-image generation with a regularization set. The authors construct this set by using ancestral sampling with formatted prompts. They tested their newly proposed regularization set on Stable Diffusion based models and achieved improvement compared to baseline.

**Strengths:**

The authors show qualitative and quantitative improvements compared to baselines in their experiments. In their qualitative examples, we can also observe that the fine-grained details of the objects are preserved better than the baselines.

**Weaknesses:**

1. It is very difficult to convince myself that the novelty presented in this paper is significant enough to warrant an acceptance. The main contribution of this paper is to construct a regularization set using a predefined and handcrafted format for prompting and ChatGPT for picking the phrases to fit the format, and the details of the format are not very well justified.

2. Continuing from Weakness 1, it is unclear to me how the authors choose the format described in Section 3 “Generating Against Training Prompts” and “Amplifying Diversity with Structured Prompts” since there is no related literature or ablation study to justify the effectiveness of each component in the format.

3. The additional time required for generating the regularization set is more substantial (2000 images for this setting v.s. < 1000 images for the original DreamBooth setting).

4. There is no evaluation on the fidelity (e.g. FID score) of the generated image, and fidelity score is a standard metric for this task.

**Questions:**

Does the size of the regularization set affect the performance? (e.g. will a smaller regularization set also work? Can the authors provide more ablation studies on this?)

---

> ### Author Response · Authors · 2023-11-23
> **Official Rebuttal**
>
> Thank you for the helpful comments. We address each question below.
>
> ## W1: Trivial contribution?
> We kindly refer you to the common question #2.
>
> ## W2: Justification of the proposed prompt format?
> In Section 4.2, we already ablate the importance of the prompt format. Without our format, the model tends to overfit to the training examples. Our format “a [adjective] [subject] [in some background]” is a simple and straight extension of the commonly used prompt “a [subject]”. We believe this simple yet effective method could inspire future researchers to advance this direction further.
>
> ## W3: Requires too many images for regularization?
> We kindly refer you to the common question #1 in the common response. According to our ablation tests, it only needs < 500 images in practice for simple cases, such as dogs and backpacks.
>
> ## W4: No evaluation on FID, which is standard for this task?
> We would like to kindly point out that FID is not a standard metric for the task of diffusion model personalization. We checked the 8 most related methods  [1,2,3,4,5,6,7,8], including the two pioneers, TextInversion [1] and DreamBooth [2]. None of them uses FID, except CustomDiffusion [3], which uses FID only to check whether the model can still generate other subjects, instead of quantifying the fidelity. In fact, in this task, researchers usually split the fidelity scores into subject fidelity and prompt fidelity: quantified with (DINO score and CLIP-I score) and CLIP-T score, respectively; which are already included in Table 1. FID is not directly applicable since it computes similarity of distributions in feature space, typically requiring thousands of reference images. However, here only a handful of example images of the target image are available. With such a low number, the FID score would be unreliable.
>
> ## Q: Does the size of the regularization set affect the performance?
> We kindly refer you to the question #1 in the common response. A smaller regularization set (<500) also works.
>
>
> [1] An Image is Worth One Word: Personalizing Text-to-Image Generation using Textual Inversion
>
> [2] DreamBooth: Fine Tuning Text-to-Image Diffusion Models for Subject-Driven Generation
>
> [3] Multi-Concept Customization of Text-to-Image Diffusion
>
> [4] Cones: Concept Neurons in Diffusion Models for Customized Generation
>
> [5] SVDiff: Compact Parameter Space for Diffusion Fine-Tuning
>
> [6] StyleDrop: Text-to-Image Generation in Any Style
>
> [7] Key-Locked Rank One Editing for Text-to-Image Personalization
>
> [8] Subject-driven Text-to-Image Generation via Apprenticeship Learning

---

### Author Response · Authors · 2023-11-23
**Response to All Reviewers**

We sincerely thank all reviewers for their valuable time and detailed reviews.

We addressed the suggested changes in the revised manuscript, all are marked blue. The main points are:
1. Additional ablation tests on the number of training examples and regularization set size in Appendix D and Figure 8.
2. Human evaluation in Section 4.1, Appendix C, and Table 2.
3. Additional qualitative comparison in Figure 13-15.
4. Additional related work to Section 2 Related Work.
5. Further automation by using BLIB to generate image-specific prompts in Appendix E.
6. To showcase the generality and robustness, we added additional 750 uncurated triplets (ours(SDXL), ours(SD), DreamBooth) in the supplementary materials “triplets.zip”.

We address the recurring questions here, and the other open questions below in response to each reviewer.

## Q1: Ablation tests on number of training examples and regularization set size (Reviewer vEAP, hHDA, AXFR)?
This is indeed an important ablation test that we did at early stages of the project but had not repeated for the full model. We now added experiments on the number of training examples (1, 2, or 4), and the number of regularization examples (100, 500, 1000, or 2000. The results are illustrated in Figure 8 (updated manuscript). To our own surprise, our full method is very robust to smaller regularization set size, as only 100 regularization examples still effectively prevent overfitting and the model is still able to preserve the very fine details of the subjects. Interestingly, when the subject is complex, only one or two training examples in combination with a large number of regularization examples (>=500) may result in underfitting, such as the backpack in Figure 8 bottom. This is not observed when the subject is simple, such as the dog in Figure 8 top. Hence, the regularization set size chosen in the main paper is conservative and works for all experiments and complexities. However, this experiment shows that  our method works with fewer regularization examples (<500) for simple cases, such as dogs and backpacks.

## Q2: Contribution is trivial (Reviewer vEAP, hHDA)?
We believe our method is simple yet effective and highly relevant. While most researchers focus on model architecture design, moving away from the data generation in DreamBooth, we propose an almost overlooked  perspective: how to prepare better training data using the word and sentence-level statistics learned by an LLM. Utilizing LLMs for improved image generation is an emergent field, e.g., OpenAI adopted recaptioning in their latest text-to-image model DALL·E 3: https://cdn.openai.com/papers/dall-e-3.pdf. Please note that their technical report was released after the ICLR deadline.

## Q3: Human evaluation (Reviewer hHDA, AXFR)?
We add human evaluation in Table 2 (updated manuscript). We asked 20 people to answer 30 questions about subject alignment and 30 questions about text alignment, in a total of 1200 questions in comparison to DreamBooth. The questions are randomly sampled from a large pool of DreamBench test prompts. The details can be found in Appendix C (updated manuscript). As shown in Table 2, our method is overwhelmingly preferred for subject alignment (64.9% / 35.1%) and similarly preferred for text alignment (52.8% / 47.2%), demonstrating the significance of the improvement.

---

### Meta-Review · Area_Chair_5nqX · 2023-12-06

**Metareview:**

(a) Scientific claims and findings

The authors introduce a systematic method for augmenting the training data for Dreambooth. The process involves enhancing training prompts with background information and generating new images from adapted prompts derived from the training set. The authors demonstrate that the augmented training data enable longer Dreambooth training and result in higher-quality output.

(b) Strengths

i. Introduce a new data-centric perspective to enhance personalized text-to-image generation.

ii. Promising quantitative and qualitative results. In particular, the proposed method enables fine-grained detail preservation in the generated images.

(c) Weakness

i. The proposed method adds a considerable computational burden to the already costly Dreambooth. The remains unaddressed and undiscussed in the paper.

ii. The paper's presentation could be improved, as it's challenging to follow. The description of the proposed approach lacks clarity for reproducibility, and it's unclear how the design choices were made and justified.

iii. The experiments are constrained to older methods and fail to include comparisons with more recent approaches.

iv. The generalizability of the proposed approach might be constrained by the restricted range of phrases/adjectives utilized in the paper.

v. Limited novelty. The proposed approach mainly consists of augmenting training data using a handcrafted template.

**Justification For Why Not Higher Score:**

There are unresolved and unaddressed concerns raised by the reviewers. These include:

1) Extensive computational overhead: This poses a significant problem in practice, yet the paper neither discusses nor addresses it effectively. The authors' response fails to adequately tackle this issue.

2) Lack of clarity in presentation: The response and revisions made by the authors do not resolve this problem.

3) Absence of comparison with more recent methods/models, including LoRA: While this may not be the primary focus of this work, the significance and contribution of the proposed method depend on the models being used.

**Justification For Why Not Lower Score:**

N/A

---

### Decision · Program_Chairs · 2024-01-16

Reject